# Preemptive Therapy in Cryptococcosis Adjusted for Outcomes

**DOI:** 10.3390/jof9060631

**Published:** 2023-05-30

**Authors:** Fernando Messina, Gabriela Santiso, Alicia Arechavala, Mercedes Romero, Roxana Depardo, Emmanuel Marin

**Affiliations:** Mycology Unit, Infectious Diseases Hospital F.J. Muñiz, Buenos Aires C1282AEN, Argentina; micologiamuniz@gmail.com (G.S.); aliarecha@hotmail.com (A.A.); mecharomero@gmail.com (M.R.); roxanadepardo@yahoo.com.ar (R.D.); emamarin@hotmail.com (E.M.)

**Keywords:** *Cryptococcus*, preemptive therapy, cryptococcal screening, lateral flow assay, cryptococcal antigenemia, serum cryptococcal antigen

## Abstract

Cryptococcosis is one of the most serious opportunistic diseases in patients living with HIV. For this reason, early diagnosis and appropriate treatment are important. Objectives. The aim of the study was to understand the development of patients diagnosed with cryptococcosis by detection of *Cryptococcus* antigen in serum by lateral flow assay (CrAg LFA) without nervous system involvement and with treatment in accordance with the results. Materials and Methods. A retrospective, longitudinal, analytical study was performed. Seventy patients with cryptococcosis initially diagnosed by serum CrAg LFA without meningeal involvement between January 2019 and April 2022 were analyzed for medical records. The treatment regimen was adapted to the results of blood culture, respiratory material, and pulmonary tomography imaging. Results. Seventy patients were included, 13 had probable pulmonary cryptococcosis, 4 had proven pulmonary cryptococcosis, 3 had fungemia, and 50 had preemptive therapy without microbiological or imaging findings compatible with cryptococcosis. Among the 50 patients with preemptive therapy, none had meningeal involvement or cryptococcosis recurrences to date. Conclusion. Preemptive therapy avoided progression to meningitis in CrAg LFA-positive patients. Preemptive therapy with dose adjustment of fluconazole in patients with the mentioned characteristics was useful despite the use of lower doses than recommended.

## 1. Introduction

Cryptococcosis is an opportunistic mycosis of worldwide distribution. It particularly affects immunocompromised patients, and is one of the main causes of morbidity and mortality in people living with HIV (PLHIV) [1,2]. For this reason, achieving early diagnosis and implementing appropriate treatment could improve the prognosis of this disease.

In 2011, the WHO recommended in a rapid advice guideline [3] the use of a quick serum antigen detection test by immunochromatography (CrAg LFA) for all patients with less than 200 CD4+ T lymphocytes/µL prior to the initiation of antiretroviral treatment. The aim of this recommendation was to avoid subsequent immune reconstitution or unmasking in asymptomatic patients.

The high sensitivity of this test in serum was proven in several studies [4,5]. For this reason, it has been recommended that this disease be diagnosed in its early stages, thus improving its prognosis.

Different economic circumstances, infrastructure, lack of trained personnel, or lack of medication mean that, in many places, different screening and early therapy schemes are used [6,7]. The optimal treatment for patients with positive CrAg LFA is unknown [8]. It is likely that each institution, given its different potential to act, should design and implement a strategy that is within its reach, with the aim of improving the morbimortality of this mycosis in this group of patients.

The current recommendations propose that CNS involvement should be ruled out and early treatment should be employed [9] in those patients in which serum cryptococcal antigen is detected by LFA. In hospitals where CrAg LFA cannot be performed, primary prophylaxis with fluconazole is recommended for all PLHIV with CD4+ T-cell counts (LTCD4+) < 200 cells/µL.

The respiratory tract is the main entry site for the infection. Most primo infections are self-limited, as a result of the immune response. In cases in which the immune system achieves control of the infection, granulomas with yeasts are generated inside of the macrophages. This state is known as latent infection, is asymptomatic, and is the main source of secondary reactivations [10]. It occurs in patients who have had an asymptomatic primary infection and then acquire a disease that impairs their immune system. In these cases, disseminated disease is usually observed as in PLHIV with low TCD4+ lymphocyte counts or patients treated with chemotherapy, immunomodulators, or corticosteroids [10].

It is known that meningeal cryptococcosis (MC) is the most frequent presentation; however, this fungus can involve various organs such as the respiratory tract, peripheral lymph nodes, bone marrow, and skin, among others [11].

For this reason, it is important, in cases with positive CrAg LFA, to perform an exhaustive semiological examination of the skin, take samples for blood cultures, perform mycological examination of CSF, chest tomography, and eventually mycological examination of respiratory material, in order to evaluate the clinical situation of the patient and thus indicate the correct treatment.

Objective. To understand the development of PLHIV hospitalized with a diagnosis of cryptococcosis by detection of CrAg LFA without central nervous system involvement and with early treatment adjusted according to the results (clinical examination, microbiological, and imaging studies).

## 2. Materials and Methods

A retrospective, longitudinal, analytical study was performed. The medical records of 203 patients with cryptococcosis diagnosed at the Mycology Unit of the F. J. Muñiz Infectious Diseases Hospital between January 2019 and April 2022 were analyzed. (Figure 1).

### 2.1. Inclusion Criteria

All PLHIV hospitalized with initial diagnosis of cryptococcosis by serum CrAg LFA without meningeal involvement. All patients aged 18 years or older.

### 2.2. Exclusion Criteria

Patients with recurrent cryptococcosis due to treatment or prophylaxis interruption. All the patients whose diagnosis of cryptococcosis was initially made by blood culture, sputum, bronchoalveolar lavage, scarification of skin or mucosal lesions, or deep biopsies. Patients who suffered meningeal cryptococcosis (MC) or who received antifungals in the 30 days previous to hospital admission. HIV-negative individuals and patients referred from other hospitals or health centers.

### 2.3. Samples

CSF: All samples were processed at the Mycology Unit according to its procedure manual [12,13].

All CSF samples were centrifuged and before processing the material 0.5 mL of the supernatant was aseptically separated to carry out antigen detection. A drop of sediment was mixed with Indian ink for direct microscopic examination (100–400×). The rest of the material was seeded on Sabouraud-dextrose agar, sunflower-seed agar, both incubated at 28 °C, and brain-heart infusion agar (BHI-agar) at 37 °C. Every culture tube was observed daily, and they were maintained for 14 days before discarding as negative.

Blood culture: blood samples were taken for blood cultures by the lysis-centrifugation method according to the technique developed in the Mycology Unit [14]. Briefly, 10 mL of peripheral blood were drawn by puncture and placed in sterile plastic tubes containing 1 mL of a saponin 5% and 0.4% of sodium salt polyanethol sulfonic acid sterile solution. After mixing several times by inversion to ensure homogenization, the content was maintained at room temperature for 1 h, then the tubes were centrifuged for 30 min, at 3000 rpm. After discarding the supernatant, sediment was seeded on Sabouraud-honey agar (at 28 °C) and brain-heart infusion agar (at 37 °C) for 3 weeks. Tubes were controlled twice a week [14].

Other materials: in patients with respiratory involvement, bronchoalveolar lavage (BAL) or sputum samples were taken. In cases with cutaneous-mucosal lesions, scarifications or biopsies were performed. Other materials such as ascitic fluid and bone marrow aspiration were also analyzed. All these samples were processed with the usual methodology for mycological diagnosis [15].

Direct microscopic examination of all these samples was performed by wet mount preparations and Giemsa stain. Cultures on Sabouraud-agar, sunflower-seed agar, and BHI-agar at 28 °C and 37 °C were maintained for two weeks.

Identification of *Cryptococcus* isolates: Every yeast colony compatible with *Cryptococcus* was typified by phenoloxidase production on Niger seed agar, urease on Christensen medium, grown capacity at 37 °C. Phenotypical differentiation between *Cryptococcus neoformans* and *Cryptococcus gattii* was carried out by seeding on glycine-canavanin-bromothimol blue agar (GCB) and glycinecycloheximide-phenol red agar (Salkin medium) [13].

Molecular identification through PCR-RFLP of URA5 gen was performed according to the following procedure:Isolation of fungal DNA (extraction by physical heat/cold treatment and subsequent treatment with proteinase K and CTAB (hexa-decyl trimethylammonium bromide) for preparation of the DNA library, which was kept at −20 °C.Amplification of Ura 5 gene by end-point PCR.Detection of the amplicon obtained by electrophoretic in 1% agarose gel.RFLP of Ura 5 with Sau96I and HhaI restriction enzymes.Detection the products with restriction enzymes on 3% agarose gel.The RFLP patterns were assigned by comparison with the patterns obtained from reference strains (*C. neoformans* var. *grubii*: CBS 10085 VNI and CBS 10084 VNII; from *C. neoformans* hybrid AD: CBS 10080 VNIII; from *C. neoformans* var. *neoformans*: CBS 10079 VNIV; and from *C. gattii*: CBS 10078 VGI; CBS 10082 VGII; CBS 10081 VGIII and CBS 10101 VGIV) [16].

Minimal inhibitory concentration (MIC) by means of the broth microdilution technique according to M27 4th Edition and M27S4 documents of the Clinical Laboratory Standard Institute–USA, was assessed to study the antifungal susceptibility of *Cryptococcus* isolates to amphotericin B (AMB) (Sigma–Aldrich, MO, USA) and fluconazole (FCZ) (Panalab, Argentine). As clinical breakpoints for *Cryptococcus* are not still determined, the epidemiological cutoffs (ECV) were used as reference to differentiate between wild-type and non-wild-type isolates [17,18,19,20].

Detection of *Cryptococcus* capsular polysaccharide antigen (CPA): detection of this antigen in serum was performed by immunochromatography (lateral flow assay IMMY, Norman Kew Surrey, Oklahoma, USA) [21].

In our hospital, this determination is performed in patients admitted to the hospital wards with LTCD4+ < 200 cells/µL or with an opportunistic disease. Semi-quantitative detection of CPA in serum and CSF is then performed using the latex agglutination (LA) technique (CryptoLatex, Immy, Norman Kew Surrey, Oklahoma, USA). To determine the titer, samples were used undiluted and in dilutions of 1:10; 1:100; 1:1000; 1:5000, and 1:10,000 [11,12].

Patients with inclusion criteria and with positive serum CRAg LFA were subjected to lumbar puncture, after neurological evaluation, to determine whether they had meningoencephalic involvement. In addition, blood culture and chest tomography were performed. Patients with respiratory symptoms or CT images compatible with pulmonary cryptococcosis were required to undergo sputum mycological examination and/or bronchoalveolar lavage.

The therapeutic regimen was adapted to the results of blood culture, respiratory material, or pulmonary tomography imaging. Patients with proven or probable fungemia and/or pulmonary cryptococcosis (PC) were medicated with two weeks of fluconazole 800 mg/day, then continued for a further 8 weeks with fluconazole 400 mg/day (consolidation) and after that switched to secondary prophylaxis or maintenance therapy with fluconazole 200 mg/day.

Patients who suffered hemodynamic decompensation or respiratory failure were treated with liposomal amphotericin B 3–5 mg/kg/day plus fluconazole 800 mg/day. Patients without pulmonary involvement and negative blood culture after 2 weeks of fluconazole 800 mg/day continued with fluconazole 200 mg/day (secondary prophylaxis or maintenance) (Figure 2).

Patients with positive serum CrAg LFA and pulmonary image compatible with PC, and for whom direct we performed examination of respiratory material with capsulated yeasts and/or culture with development of *Cryptococcus neoformans* or *Cryptococcus gattii* were considered as proven PC.

Patients with positive serum CrAg LFA and pulmonary image compatible with PC were considered probable PC.

Patients with PC or fungemia who were also receiving rifampicin (a potent inducer of cytochrome P450) due to tuberculosis, received fluconazole 800 mg/day during the entire consolidation period.

Patients who were due to switch to prophylaxis but were also receiving rifampicin or for whom the blood culture was performed late (after starting fluconazole) completed the 8-week regimen with fluconazole 400 mg/day (prolonged consolidation therapy). In some patients with tuberculosis, rifampicin was changed to another medication.

### 2.4. Statistical Analysis

Data for continuous variables were expressed as mean or median when applicable. The categorical variables were expressed as frequencies. The Fisher’s exact test and the chi-square test were used to show statistical differences between groups. A *p*-value of <0.05 was considered significant. Statistix^®^ 8.0 software was used for the analysis.

## 3. Results

In the period analyzed, 1698 serum CrAg LFA tests were performed in hospitalized PLHIV. One hundred ninety-nine were positive (11.7%). Considering only the cases with extrameningeal cryptococcosis, this methodology made possible the diagnosis in 4.1% of the patients hospitalized in that period (70 patients).

Seventy patients met the inclusion criteria; 39 were male (55.7%), the median age was 40 years (range: 22–69 years), the median LTCD4+ was 35 cells/µL (range: 3–162 cells/µL). The most frequent comorbidity with significant differences was tuberculosis (TB) (17 patients, *p* = 0.0002), followed by cerebral toxoplasmosis (7), pneumocystosis (6), bacterial pneumonia (5), hepatitis C (4), COVID-19 (2), and disseminated CMV (2).

The two most frequent symptoms at admission with significant difference were fever and impregnation syndrome (54 and 42 patients, respectively, *p* < 0.0001), followed by cough (24), dyspnea (15), diarrhea (10), and motor focus (7).

The mycological examination findings, imaging, and clinical presentations with the different treatments and the evolution can be seen in Table 1.

There were no evident lesions in the brain CT scan or even in magnetic resonance imaging in many cases. The opening pressure was within normal parameters in all cases.

Four patients had proven pulmonary cryptococcosis (PC) (Figure 3a), thirteen had probable PC, three had cryptococcal fungemia, and 50 continued with early therapy. Of the latter, six patients had to undergo prolonged consolidation therapy.

In relation to the pulmonary cryptococcosis tested, in three cases the development of this fungus was obtained in the culture. In only one case it was visualized in the direct examination in fresh and with the addition of Indian ink to the respiratory material to evidence the characteristic capsule. In this last case, the fungus did not develop on the usual culture media.

All the Cryptococcus isolates belonged to C. neoformans complex, and their genotype was VNI.

Susceptibility tests showed that the AMB was tested onsix C. neoformans isolates, and all showed a MIC value ≤1 µg/mL. Susceptibility to FCZ was determined in six isolates, and was ≤8 µg/mL. All the isolates presented MICs lower than the ECV (95% ECV 16 µg/mL; 99% ECV 32 µg/mL), for both drugs.

Two patients with probable PC were treated during consolidation with fluconazole 800 mg/day due to concomitant pulmonary tuberculosis.

Two patients died. One of them had proven severe PC and pneumocystosis [22] (Figure 3b), the other during secondary prophylaxis, after completing the probable PC scheme, due to a sarcoma in the pelvic region. The rest of the patients progressed favorably. All patients who received pre-emptive therapy had a favorable outcome and after 12 to 52 months (depending on the time of diagnosis), none of them had subsequent meningeal involvement. Antiretroviral treatment (ART) was indicated for every patient, assessing the context and situation of each one. Once meningeal involvement was excluded, 43 patients started ART before the first two weeks, 15 before three weeks and 11 patients four weeks after starting antifungal treatment.

## 4. Discussion

Cryptococcal infection is increasingly recognized as a major cause of invasive fungal condition not only in PLHIV, also in solid organ transplant recipients. However, in solid organ transplant recipients, a study of more than 1000 patients identified cryptococcal infection as the third most frequent fungal pathology after those caused by *Candida* and *Aspergillus* species. The limited use of strategies to diagnose cryptococcal disease early, before the onset of meningitis, contributes to increased morbidity and mortality, particularly in resource-poor settings. Diagnostic tests have been used for several years to identify patients with cryptococcosis. These include the latex agglutination (LA) test and enzyme immunoassay (EIA). However, in resource-limited settings, both are difficult to use, as they require refrigeration, pre-treatment with additional enzymes, such as pronase in the case of LA, and specialized equipment, such as a spectrophotometer in the case of EIA [23].

In 2009, a new lateral flow assay (LFA) was developed that allows rapid detection of the cryptococcal glucuronoxylomannan polysaccharide capsule, and in 2011 [3] it was approved by the FDA for use in primary care centers because of its ability to detect cryptococcal capsular antigen from all serotypes, its high sensitivity, and the fact that it provides results in 10–15 min without the need for highly specialized personnel for its performance. Its usefulness in revealing disease in asymptomatic PLHIV was later established. The use of random-effects meta-analysis made it possible to analyze the overall sensitivity and specificity in both serum and CSF. The sensitivity in serum was 99.8% and in CSF 98.8% while the specificity was 95.2% and 99.3%, respectively. In addition, the concordance between culture and CrAg-CSF was 97% [24]. The test provides qualitative and semi-quantitative results within ten minutes without sample pretreatment. The test materials do not require refrigeration or cold chain transport. This allows it to be used in health centers with a smaller infrastructure than other centers [25].

To measure fungal burden, semi-quantitative studies can be done with serum CrAg titers using either LFA or LA, but it should be noted that LFA is 4–5 times more sensitive.

As with quantitative cultures, antigen titers can predict progression to meningitis, and when antigenemia titers by LFA are >160 the patient is at increased risk of mortality [26]. On the other hand, patients with serum/plasma CrAg titers ≤ 1:80 have reduced probability of CNS involvement [27,28,29].

Molecular methods have made it possible to understand the genetic differences between the various *Cryptococcus* species and to recognize which genetic factors contribute to their pathogenicity and virulence. As in most parts of Latin America, the most frequent genotype in this study was VNI [30].

Firacative also published results indicating that the majority of 570 Latin American isolates proved to be wild-type against fluconazole, amphotericin B, itraconazole, voriconazole, 5-fluorocytosine, and posaconazole. However, non-wild-type VNI strains were isolated for fluconazole and amphotericin B in Brazil and Argentina [2]. In our study all the isolates were wild-type against fluconazole and amphotericin B.

Preemptive therapy is defined as the indicated treatment with fluconazole in high-risk, asymptomatic patients with positive serum LFA CrAg.

The use of serum LFA CrAg to detect patients with cryptococcosis was initially recommended in countries or cities with a prevalence of the disease greater than 3% [9]. However, this methodology is also useful in cities with lower prevalence [31].

The first cost-effectiveness studies carried out on the use of early therapy with fluconazole vs. without antifungal treatment showed that it increased survival [28]. It has been observed that in asymptomatic patients the use of fluconazole alone is often not sufficient [8].

Despite the use of the screening methodology recommended by the WHO with subsequent anticipatory therapy, some authors continue to report high mortality. This is probably due to the fact that many cases with meningeal or other organ involvement are treated with fluconazole alone [32,33]. Screening with all possible samples for mycological examination is essential to indicate the appropriate treatment. In addition to taking clinical samples, imaging studies are also very important. Among the latter, the thoracic tomography seems to be very important to show images that are not usually seen in a simple thoracic radiography. Despite evidence that screening with CrAg reduces the incidence of cryptococcal meningitis and related mortality, there remains a persistent association between CrAg positivity and death.

We should look for and confirm the clinical presentation of the patient once the result of a positive CrAg LFA is available and before starting any treatments [32]. Any organ can be affected after hematogenous dissemination. Once a patient presents with a positive CrAg LFA antigenemia and before initiating treatment, it is necessary to confirm the etiologic diagnosis of the condition and which organs or systems are involved [32].

In immunosuppressed individuals with positive serum CrAg LFA, pulmonary cryptococcosis or in other localizations, especially meningitis, should be ruled out by CSF mycological study, since the presence of CNS involvement alters the dose and duration of induction therapy and the need to monitor intracranial pressure [31]. It has been widely demonstrated that therapy with fluconazole as the only drug in cases of meningitis is associated with high mortality [32]. For this reason, when lumbar puncture is not possible, serum CrAg titers can help to decide between the use of two drugs in induction or monotherapy [32,34], because patients with serum titers higher than 1/160 by LFA or 1/100 by LA have a higher probability of meningitis [34,35]. Consequently, mortality is higher in those patients [36]. Cryptococcosis is a fungal infection that requires a rapid and adequate diagnosis in order to shorten hospitalization times and achieve a favorable development.

In cases in which the diagnosis is made through serum CrAg LFA without target organ involvement, early therapy and subsequent secondary prophylaxis is employed in order to avoid meningitis, immune reconstitution syndrome, or unmasking when antiretroviral therapy is indicated.

Originally, LFA CrAg was recommended in patients with LTCD4+ < 100 cells/µL, but today it is also used in patients with LTCD4+ < 200 cells/µL [32]. At the beginning it was recommended that patients with positive serum LFA CrAg should undergo a lumbar puncture to rule out meningeal cryptococcosis (MC) only if the patient had symptoms. In asymptomatic patients, early therapy with fluconazole was initiated to prevent future CM. Recently, it has been observed that many asymptomatic patients may have CM [26,37]. Therefore, lumbar puncture should be performed in all patients with positive serum LFA CrAg.

As we observed in this group of patients, the symptoms that lead to consultation are very unspecific, so it is necessary to perform this determination in all patients who are hospitalized with less than 200 cells/µL LTCD4+ or with suspicion of opportunistic disease.

Non-contrast chest CT is essential because many of the patients have pulmonary nodules of probable fungal etiology and no respiratory symptoms. The most frequently observed pulmonary images in PC are interstitial pneumonitis and pulmonary nodules [38], in some cases cavitated in the center.

Some authors, once they exclude meningeal involvement, have considered starting treatment with fluconazole 1200 mg/day for the first two weeks [39], in other cases they initially used 900 mg/day [40], and in others the consolidation therapy was shorter [41] than indicated in the guidelines [9]. A recent study carried out in Brazil showed a prevalence with no differences in the prevalence of CrAg stratified LTCD4 values (CD4 < 100 vs. CD4 101–199 cells/μL) [40]. In that study, no clinical or laboratory factor predicted CrAg positivity, which corroborates the need to implement universal screening for CrAg in PLHIV with CD4 <200 cells/μL in similar settings. Borges M et al. employed pre-emptive therapy with fluconazole 900 mg/day for two weeks and then 450 mg for 8 to 10 weeks and a subsequent maintenance dose of 150–300 mg [40]. As in our study, they had no relapse of cryptococcosis after 12 months. On the other hand, a study that evaluated patients screened with CrAg LFA with subsequent pre-emptive therapy with fluconazole, in which meningeal involvement was ruled out but other focuses were not well evaluated, showed a mortality rate close to 25% despite the treatment with fluconazole in early therapy [32].

To date, there are no clinical trials for cases of cryptococcosis in which there is no pulmonary or central nervous system involvement. Therefore, there are differences in the way to treat with antifungals when the CrAg LFA is positive and there is no involvement of the meninges, lungs, or other organs.

Patients with mild pulmonary disease have been successfully treated with fluconazole monotherapy 400 mg/d [42] and in severe cases combined treatment with liposomal amphotericin B or deoxycholate combined with flucytosine or fluconazole is recommended [42].

The heterogeneity of the treatments and complementary studies used in patients diagnosed with serum CrAg LFA+ is probably related to the diversity of diagnostic and economic possibilities.

In our cases, it was possible to adapt the treatments to the results obtained by the different chest images and clinical samples analyzed.

Reducing the dose of fluconazole in cases without pathological findings is likely to improve patient adherence to the different treatments.

In patients suffering from CM it is established that ART should be started 4–6 weeks after initiation of antifungal therapy [43]. However, the initiation of antiretrovirals in patients without CM and with other forms of cryptococcosis has not yet been established.

Identifying the different clinical presentations is essential in order to initiate an early therapy properly adjusted to the results obtained. It is very important that microbiological studies or mycological examinations are performed prior to initiating therapy with fluconazole. Interactions of this antifungal drug with other medications should be evaluated on a case-by-case basis. It should be remembered that tuberculosis is a very frequent disease in our environment, especially in PLHIV. In that disease, rifampicin is an essential drug for the good development of the patients. This drug is an inductor of the cytochrome P450 system at the hepatic level (a potent inducer of CYP3A4) and therefore accelerates the metabolism of many drugs that share this metabolic pathway, causing a decrease in their plasma concentrations. Fluconazole is metabolized by CYP3A4, therefore the use of both drugs causes lower fluconazole concentrations in peripheral blood. This could lead to subtherapeutic doses of fluconazole, resulting in treatment failure or the possibility of fluconazole resistance if used at low doses.

It remains important to remember that although our study is based on hospitalized patients with suspected opportunistic disease, we should always evaluate the possibility of performing LFA in all patients with less than 200 LTCD4 before initiating antiretroviral therapy (ART).

Early initiation of ART in patients with CM can lead to life-threatening immune reconstitution syndrome (IRIS), so delaying ART is recommended.

However, many cases of CM and IRIS that occur after initiation of antiretroviral therapy could be prevented [6].

In addition, it should be considered that the increase of cryptococcosis cases in non-HIV immunocompromised patients (solid organ transplant recipients, autoimmune diseases, patients under treatment with corticosteroids or immunomodulators, idiopathic immunodeficiencies, hematologic diseases, etc.) requires the use of rapid diagnostic techniques such as LFA CrAg in serum to detect this pathology early and to be able to indicate the appropriate treatment.

It is important to mention that this study has the limitation of being a retrospective study of a single center and with a limited number of patients from Argentina. For this reason, further studies are needed to confirm the findings of our work.

## 5. Conclusions

In patients with positive serum CrAg LFA, it is essential to evaluate the clinical situation of the patient, with a thorough semiology examination, request lumbar puncture to rule out central nervous system involvement, perform blood cultures and thoracic CT scan with possible mycological examination of respiratory material, and then evaluate the best treatment based on the results.

Finally, this treatment with dose reduction of fluconazole in patients in whom the mentioned focuses were ruled out was useful even with doses lower than those recommended.

## Figures and Tables

**Figure 1 jof-09-00631-f001:**
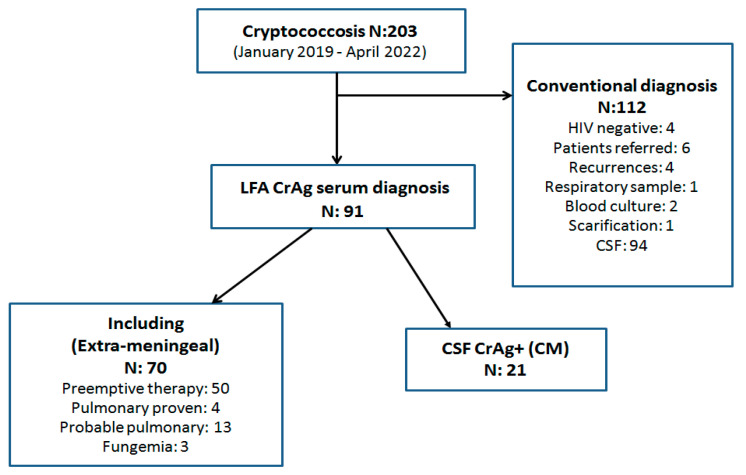
Flow chart with patients included and final results.

**Figure 2 jof-09-00631-f002:**
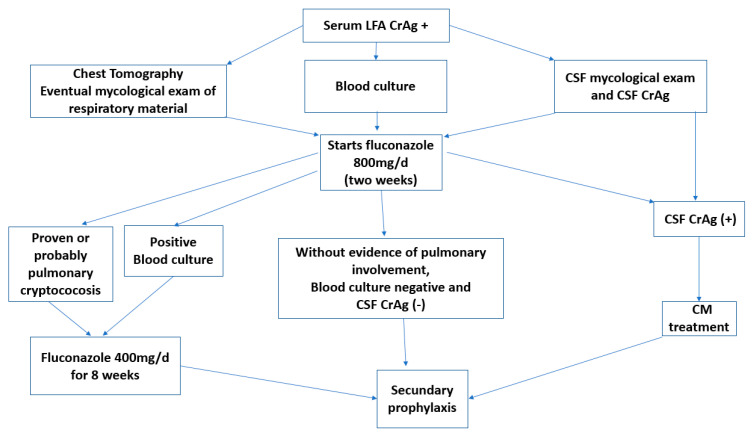
Methodologies used for diagnosis and treatment.

**Figure 3 jof-09-00631-f003:**
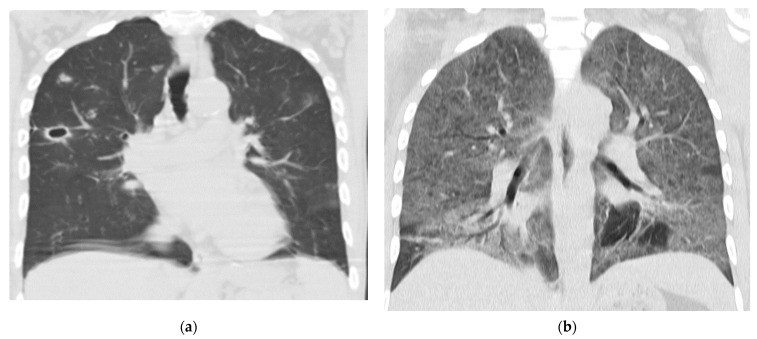
(**a**) Small thin-walled cavity and pulmonary nodules in patient with proven CP. (**b**) Chest CT scan, coronal image showing interstitial pattern with ground-glass view.

**Table 1 jof-09-00631-t001:** Clinical and microbiological characteristics related to the treatment and evolution of the different groups.

		Preemtive Therapy: N:50 (%)	Probable Pulmonary: N:13 (%)	PulmonaryProven: N:4 (%)	Fungemia: N:3 (%)
	Median age	42	40	35	36
	Male gender	30	5	2	2
	Median LTCD4+	48	72	22	27
Earlysymptoms	Impregnation syndrome	26 (52)	9 (69)	4 (100)	3 (100)
	Fever	35 (70)	12 (92)	4 (100)	3 (100)
	cough	17 (34)	6 (46)	1 (25)	0
	dyspnea	9 (18)	2 (15)	3 (75)	1 (33)
Pulmonary pattern	Interstitial pattern	3 (6)	2 (15)	3 (75)	1 (33)
	Pulmonary nodules	0	11 (85)	1 (25)	0
Respiratory material	Positive microscopy	0	0	1 (25)	0
	Positive Culture	0	0	3 (75)	0
Comorbidity	tuberculosis	16 (32)	0	0	1 (33)
	toxoplasmosis	6 (12)	1 (8)	0	0
	pneumocystosis	3 (6)	1 (8)	2 (50)	0
Treatment	fluconazol 800 mg (2 weeks) then 400 mg (8 weeks) then 200 mg *	6 (12)	11 (85)	3 (75)	2 (66)
	fluconazol 800 mg (10 weeks) then 200 mg *	0	2 (15)	0	1 (33)
	fluconazol 800 mg (2 weeks) then 200 mg *	44 (88)	0	0	0
	Amphotericin B + fluconazole 800 mg	0	0	1 (25)	0
	Death	0	1 (8)	1 (25)	0

* Secondary prophylaxis.

## Data Availability

All data generated or analyzed during this study are included in this published article.

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
