# Peer review of "Preemptive Therapy in Cryptococcosis Adjusted for Outcomes"

_jof, 2023, doi:10.3390/jof9060631_

Round 1
Reviewer 1 Report
The article by Messina et al. aims at assessing the outcome of patients with detection of Cryptococcus antigen in serum by lateral flow assay and adapted early treatment.
The issue is of interest for the readers of the Journal of Fungi, but my main concern is that the presentation of the results does not make the readers appreciate well what the study brings to this issue. To my mind, the authors should present a table summarizing patient clinical, biological and radiological presentation combined with treatment modalities, and examine the association with the outcome and duration of evolution.
Minor comments:
Introduction, lines 50-52: “In hospitals where CrAg LFA cannot be performed, primary prophylaxis with fluconazole is recommended for all PLHIV with CD4+ T-cell counts (LTCD4+) < 200 cells/μL.” Don’t you recommend this prophylaxis event in a hospital where CrAG can be performed?
Materials and methods, lines 97-98 ans 115: “they were maintained for 14 days”. Cryptococcus growth can be delayed, so why were cultures not prolonged during 3 weeks as recommended?
Line 102: what do you mean with “0.4 venous %”?
Lines 164-165: “Patients with hemodynamic decompensation or respiratory failure, treatment with liposomal amphotericin B 3-5 mg/kg/day plus fluconazole 800 mg/day was indicated.” Please mind the syntax.
Line 224: “nearly all”. Please specify.
Discussion, lines 247-249: “The limited use of strategies to diagnose cryptococcal disease early, before the onset of meningitis, contributes to decreased morbidity and mortality, particularly in resource-poor settings.” The sentence is confusing as this limited use would rather contribute to increase morbidity and mortality.
Lines 255-259: a reference is missing.
Line 271: “when antigenemia titers by LFA are >160 the patient is at increased risk of mortality.” Please add the reference: Wake CID 2018.
Lines 281-282: “In our study all the isolates were wild-type against fluconazole and amphotericin B [2].” Reference 2 should be written at the end of the previous sentence.
Lines 343-345: “A recent study carried out in Brazil showed a prevalence with no differences in the prevalence of CrAg stratified LTCD4 values (CD4 <100 vs. CD4 101-199 cells/μL).” A reference is missing.
Line 347: “Borges et al.” Please add reference 41.
Abstract, line 12: “lateral flow” should read “lateral flow assay”
Materials and methods, lines 79-89: please avoid telegraphic style.
Figure 2: prophilaxis should read prophylaxis
Results, line 208: please mind the syntax.
Line 256: glucuoronoxylomannan should read glucuronoxylomannan.
Line 258: sensibility should read sensitivity.
Line 346: “PLWHA”. This abbreviation has not been explained. Please homogenize the use of PLHIV or PLWHA.
Line 369: “In patients suffering CM” should read “In patients suffering from CM”.
Lines 386-388: “We should always evaluate the possibility of performing LFA in all 380 patients with less than 200 LTCD4 before initiating antiretroviral therapy (ART).” This sentence has been written twice.
Author Response
Dear reviewer: thank you very much for your comments and suggentions. They have greatly improved the quality of the article.
I send the answer to each of the points as an attachment

Reviewer 2 Report
This study assessed the effect of pre-emptive therapy for cryptococcosis and showed the clinical benefit of this intervention. This study is well-designed. I just have several minor suggestions.
1. Please discuss the limitation of the study, such as single center, or small case number.
2. Figure 3 can be deleted.
3. Please add new table to summarize the characteristics of the included patients and briefly describe in the result section.
Author Response
Dear reviewer: thanks for your comments. I send the answer to each of the points as an attachment

Reviewer 3 Report
The manuscript addresses a subject previously analyzed in other regions, so, the novelty is somehow compromised. Few studies about cryptococcosis are reported from Argentinian Hospitals, which is of local and regional interest.
The main limitation of this study is the small number of patients that fulfilled the inclusion criteria, which undermines the power of the given observations. Regarding the exclusion criteria, it should include the use of antifungal drugs prior to the appearance of symptoms related to the clinical presentation of cryptococcosis, this would bias both identification and antifungal sensitivity profiles.
I disagree with the conclusions, there is not a single piece of evidence that supports those claims, particularly "Preemptive therapy prevented progression to meningitis in patients with positive serum CRAg LFA. Finally, this therapy with dose reduction of fluconazole in patients with the mentioned characteristics was useful even with lower doses than those recommended." This kind of statement may be elaborated after comparing groups with different therapeutic schemes or treated with different antifungal drugs, but this is not the case. The whole conclusion section has to be modified. In fact, I recommend that these results should be communicated as a letter to the editor/short note/observation. In this sense, no conclusions are required. With the small number of patients enrolled and with the given evidence, I see it hard to provide any solid conclusion.
English is fine, the manuscript needs minor polishing work.
Author Response
Dear Reviewer: thank you very much for your comments and suggestions.
I send the answers to each of the points as an attachment.

Round 2
Reviewer 1 Report
Thank you for having taken into account most of the comments. However a few issues remain.
Materials and methods, lines 96-97 and 114: I still believe that with a 14-day culture you may miss some Cryptococcus strains that can grow slowlier (as with a 21-day culture you may miss some Histoplasma strains that can grow slowlier), but I acknowledge the fact that it is not possible to change culture duration for this study.
Table 1: The table is not properly presented: please add % when appropriate. Please specify for example that the data are presented as n (%), unless otherwise indicated. Age data are usually presented as median (IQR).
Discussion, lines 247-249: The sentence is still confusing as this limited use would rather contribute to increase morbidity and mortality. I would rather write: “The limited use of strategies to diagnose cryptococcal disease early, before the onset of meningitis, contributes to increase morbidity and mortality, particularly in resource-poor settings.”
Thanks for having made the corrections.
Author Response
Thank you very much for your help
I send you the changes by attachment

Reviewer 3 Report
Thanks for addressing my concerns. The sole criticism I have of this version of the manuscript is related to the conclusion. It is not possible to conclude something that is "probably preventing". A conclusion is a solid statement coming from experimentation and anything that is probable is far away from solid. My suggestion is to remove this second paragraph from the conclusion.
Author Response
Thank you for your help
I send the changes by attachment
